# Recent Advances in Sequential Infiltration Synthesis (SIS) of Block Copolymers (BCPs)

**DOI:** 10.3390/nano11040994

**Published:** 2021-04-13

**Authors:** Eleonora Cara, Irdi Murataj, Gianluca Milano, Natascia De Leo, Luca Boarino, Federico Ferrarese Lupi

**Affiliations:** 1Advanced Materials and Life Sciences, Istituto Nazionale di Ricerca Metrologica (INRiM), Strada delle Cacce 91, 10135 Turin, Italy; irdi.murataj@polito.it (I.M.); g.milano@inrim.it (G.M.); n.deleo@inrim.it (N.D.L.); l.boarino@inrim.it (L.B.); 2Dipartimento di Scienza Applicata e Tecnologia, Politecnico di Torino, C.so Duca degli Abruzzi 24, 10129 Turin, Italy

**Keywords:** sequential infiltration synthesis, block copolymer, self-assembly

## Abstract

In the continuous downscaling of device features, the microelectronics industry is facing the intrinsic limits of conventional lithographic techniques. The development of new synthetic approaches for large-scale nanopatterned materials with enhanced performances is therefore required in the pursuit of the fabrication of next-generation devices. Self-assembled materials as block copolymers (BCPs) provide great control on the definition of nanopatterns, promising to be ideal candidates as templates for the selective incorporation of a variety of inorganic materials when combined with sequential infiltration synthesis (SIS). In this review, we report the latest advances in nanostructured inorganic materials synthesized by infiltration of self-assembled BCPs. We report a comprehensive description of the chemical and physical characterization techniques used for *in situ* studies of the process mechanism and *ex situ* measurements of the resulting properties of infiltrated polymers. Finally, emerging optical and electrical properties of such materials are discussed.

## 1. Introduction

The seek for novel materials with tailored properties has been of great interest among the scientific community over the last decades. The ability to fabricate nanostructured inorganic materials with high degree of control on morphology and dimensions, led to advanced materials with boosted performances in different research fields, such as nanolithography [1,2,3,4], photonics [5], biomedicine [6,7] and energy [8,9]. The realization of wide-area periodic nanopatterns is currently the subject of many efforts by the microelectronics industry, pushing the development of next-generation electronic and optical devices. At the moment, conventional lithographic techniques (i.e., optical and electron lithographies) represent the workhorse of micro and nanoscale manufacturing. Over the last years, their technological improvements determined significant advances, approaching the fundamental requirements demanded by the continuous downscale of device features. However, conventional lithographic techniques are now facing their intrinsic technological and economic limits [10] in terms of large-scale pattern definition and material deposition.

Among alternative nanopatterning methods, self-assembled materials such as block copolymers (BCPs) demonstrated to be very valuable in the pursuit of the shrinkage of electronic and optical devices, offering large scale scalability and a ready integration in the manufacturing processes [10,11]. The self-assembly of BCPs, in particular, represents a cost-effective bottom–up approach with high throughput, able to provide highly dense periodic patterns at the nanoscale in the typical range of 10–100 nm. Compared to optical and electron lithography, the self-assembly of BCPs relies on the in-parallel self-registration of amphiphilic BCPs, driven by the chemical incompatibility between the constituent blocks. A high degree of control on self-assembled nanostructures, in terms of orientation [12,13], long-range ordering [14,15,16], morphology [17] and feature size [18,19] is related to the ability to finely tune the substrate functionalization, annealing conditions and the characteristic parameters of BCPs (i.e., molecular weight and composition). The potential use of BCPs for several semiconductor industry technologies was recently assessed by Liu et al. [10]. By a direct comparison of directed self-assembly (DSA) of BCPs with conventional multi-step patterning approaches, such as self-aligned double/quadruple patterning (SADP/SAQP); the authors demonstrated the feasibility of applying BCP nanopatterning in the fabrication of 7 nm node fin field-effect transistors (FinFETs) in high-volume manufacturing testing. In addition, the pattern quality of fabricated patterns, in terms of critical dimension and pitch uniformity, was reported to be sufficient for integrated circuit layer manufacturing. The overall lower processing cost and high scalability provided by self-assembly of BCPs could also pave the way for the fabrication of self-assembled crossbar arrays of memristive devices for the realization of next-generation computing architectures, as also underlined in the roadmap on emerging hardware and technology for machine learning [20]. The great flexibility provided by the BCPs offers the opportunity to employ them as a nanopatterning tool for the design and fabrication of a wide range of functional materials. In particular, when combined with emerging synthetic routes as sequential infiltration synthesis (SIS), BCPs represent ideal templates for the synthesis of hybrid organic/inorganic or all-inorganic nanostructured materials with potential applications spanning from nanoelectronics [21] to photonics [22] and optics [23]. The SIS process is a vapor-phase and solvent-free process based on atomic layer deposition (ALD), generally used for the inclusion of inorganic materials into polymer templates. SIS consists of the cyclic exposure of polymers to a vapor-phase metal–organic precursor and an oxidizing agent ($H_{2}O$, $H_{2}O_{2}$, $O_{3}$), which leads to the formation of organic/inorganic hybrid materials. When SIS is applied to self-assembled BCPs, the metal–organic precursors are selectively entrapped inside the polar homopolymer composing the BCPs. Subsequent removal of the polymeric species, obtained whether by polymer ashing [24] or plasma etching [25], reveals a nanostructured metal oxide whose morphology perfectly replicates that of the BCPs template [26], as schematized in Figure 1.

Although sharing the same equipment and metal–organic precursors, the processing parameters of SIS substantially differ from that of conventional ALD processes, widely used for the conformal deposition of inorganic thin films on solid substrates (Figure 2a). Indeed, in conventional ALD, the cyclic exposures to the metal–organic precursors are typically very short, at low partial pressure and aimed at saturating all the reactive sites on the substrate surface in a self-limiting fashion. By contrast, in SIS the goal is to dissolve, diffuse and entrap the precursors throughout the entire BCPs film thickness (Figure 2b), thus requiring higher exposure partial pressures and times [27,28,29]. The extensive research over the last years has referred to SIS with different terminologies i.e., vapor phase infiltration (VPI) [30], micro-dose infiltration synthesis (MDIS) [31] and multipulse vapor infiltration (MPI) [32]. Although each process indicates a different precursor dosing sequence, they all rely on the same fundamental phenomenology [30].

Here, we report recent advances and perspectives of the SIS process, with a specific focus on the synthesis of nanostructured materials by BCPs templates. Great attention is dedicated to the discussion of *in situ* and *ex situ* spectroscopic and microscopic characterization techniques adopted for an exhaustive comprehension of the process mechanism and morphological, compositional and structural characterization. Subsequently, in this review, we address the emerging optical and electrical properties of infiltrated materials with potential technological impact on the development of novel devices.

## 2. SIS Processing and Mechanism

The SIS of BCPs follows a Lewis acid–base interaction between the metal–organic precursors (Lewis acids) and functional groups of the polar domains (Lewis bases). Being polystyrene-*block*-poly(methyl methacrylate) (PS-*b*-PMMA) the prototypical BCPs, widely used as reference material for the study of the self-assembly process, a lot of effort has been dedicated to the understanding of the mechanism involved in the SIS [25,33,34]. Early studies on the synthesis of aluminum oxide (${\mathrm{AlO}}_{x}$) obtained after the cyclic exposure of PS-*b*-PMMA to trimethylaluminum (TMA) and water, demonstrated that the TMA–PMMA interaction follows a two-step adsorption [35]. The first step consists in the formation of a Lewis adduct obtained by the reversible coordination of TMA to the carbonyl (C=O) of the ester groups of PMMA, then followed by a slow conversion into covalent Al–O bond [36]. Subsequent exposure to water determines the formation of O–Al–OH species, due to the oxidation of bonded TMA, that act as nucleation and growth sites for ${\mathrm{AlO}}_{x}$ in the following SIS cycles [31]. The lack of polar functional groups in PS implies the absence of any interaction of the precursors with the aforementioned homopolymer. Consequently, PS acts as a diffusive channel for the transport of the precursors to the reactive sites of PMMA [33]. A similar behavior is also found in the statistical copolymer polystyrene-*stat*-poly(methyl methacrylate) (PS-*stat*-PMMA). However, the TMA diffusivity is affected as the MMA unit content in the polymer film varies, reaching a maximum value for MMA fraction of 0.56 [37]. The inert properties of PS towards metal–organic precursors has been recently exploited for the uniform coating of freestanding nanoparticles. By applying the SIS on resting nanoparticles on a PS layer, the precursors can diffuse through the underlying PS and reach the reactive sites on the bottom part of the nanoparticles. This allows the growth of the metal oxide on nanoparticles even on the side in contact with the substrate, otherwise not possible with standard ALD process [38].

### 2.1. Polymer Selectivity

The search for a comprehensive insight into the SIS mechanism has also been extended to polymers with amides and carboxylic acids functional groups, such as poly(vinylpyrrolidone) (PVP) and poly(acrylic acid) (PAA), respectively. While PVP shows similar reactivity to PMMA, forming a reversible Lewis adduct $C=O\cdot \cdot \cdot \mathrm{Al}--{\left(CH_{3}\right)}_{3}$, in PAA the presence of an acidic proton determines the direct covalent Al–O bonding through a pericyclic reaction [39] (Figure 3).

Different polymers with carbonyl-containing functional groups, therefore, show substantial differences in the interaction dynamics with the metal–organic precursors. Biswas et al. [40] recently reported that, although sharing the same ester functional groups, poly(ϵ-caprolactone) (PCL) interacts more strongly with TMA and ${\mathrm{TiCl}}_{4}$ compared to PMMA, showing nearly total saturation of the available C=O sites for both precursors. The higher reactivity of PCL is to be found in the polymer backbone positioning of the carbonyl groups that confers a higher nucleophilicity compared to the side chain C=O groups in PMMA, resulting in a stronger Lewis acid–base interaction with metal–organic precursors.

The increasing research on new polymers with oxygen-containing functional groups pushes forward the achievement of direct selective growth of different nanostructured metal oxides as ZnO, ${\mathrm{TiO}}_{x}$ and ${\mathrm{VO}}_{x}$ that otherwise would require pre-infiltration of ${\mathrm{AlO}}_{x}$ [41,42,43]. As an example, Yi et al. [44] reported how cyclic ether groups of polystyrene-*block*-poly(epoxyisoprene) (PS-*b*-PIO) act as effective templates for the direct infiltration synthesis with TMA, diethylzinc (DEZ), titanium isopropoxide ($\mathrm{Ti}{\left(\mathrm{OiPr}\right)}_{4}$) and vanadyl isopropoxide ($\mathrm{VO}{\left(\mathrm{OiPr}\right)}_{3}$) thanks to a greater Lewis basicity of cyclic ether groups when compared to the ester group of PMMA.

Surprisingly, the same authors found a selective growth of ZnO and ${\mathrm{AlO}}_{x}$ in polyisoprene domains of polystyrene-*block*-poly(1,4-isoprene) (PS-*b*-PI) BCPs even though lacking any polar ligand group, suggesting that the Lewis acid–base interaction alone is insufficient to fully describe the precursor entrapment. A first attempt of explanation on how alkene functional groups in PS-*b*-PI can play a role in entrapping metal–organic precursors was given by attributing it to the high permeability of PI to a given precursor. Lately, a more in-depth assessment of the mechanism involving SIS with DEZ in cis-polyisoprene, revealed that pre-heating treatments play a key role in increasing the load of metal–organic precursors by inducing chemical changes to cis-polyisoprene. Indeed, pre-heated films undergo partial oxidation, which introduces new C=O functional groups responsible for the increased metal–organic entrapment [45]. A list of relevant references focusing on SIS on different polymers and functional groups is presented in Table 1.

The extensive literature on SIS of nanostructured metals and metal oxides as ${\mathrm{AlO}}_{x}$ [33], ${\mathrm{SiO}}_{x}$, ${\mathrm{TiO}}_{x}$ [48], ZnO [31], W [25] and ${\mathrm{WO}}_{3-x}$ [42] proved self-assembled BCPs templates as a promising tool for nanopatterning applications, thus pushing the research to the development of SIS for new semiconducting materials such as In_2_O_3_, Ga_2_O_3_ [49] and ${\mathrm{SnO}}_{x}$ [43]

### 2.2. Diffusion

When comparing the phenomena involved in SIS (i.e., sorption, diffusion and entrapment) to ALD, a higher complexity is determined by the larger number of experimental design parameters that need to be taken into account, namely: temperature, pressure, pre-treatments, precursor and oxidizing agent exposure times, purge time and polymer–precursor interaction [30]. The ability to perform deep infiltration of inorganic materials into polymers represents one of the fundamental aspects to expand the technological impact of SIS on a wide range of applications. The diffusion of inorganic precursors into polymeric templates, although being of critical importance, suffers from limitations in terms of depth of penetration that affect the inorganic material mass incorporation and pattern quality [43]. Different strategies have been recently developed in order to increase the effective diffusion of metal–organic precursors into polymer templates. Examples of infiltration of PS-*b*-P4VP (polystyrene-*block*-poly(4-vinylpyridine)) in polar swelling solvents (i.e., ethanol), show a more efficient infiltration thanks to the introduction of additional porosity channels [46]. The swelling-assisted SIS is a method based on the immersion of BCP films into a polar solvent prior to the infiltration. The incorporation of polar organic solvents into the polar domains of the BCP, upon subsequent drying, determines the formation of interconnected pores in the typical range of 10–50 nm. These pores act as effective pathways for the delivery of the metal–organic precursors throughout the BCP film thickness [50]. Thus, they enable the access of the metal–organic precursors to all the available sites. This results in a two-fold increase of the amount of synthesized ${\mathrm{AlO}}_{x}$, proving to be a valid approach also for the synthesis of porous multicomponent heterostructures [47]. Higher amounts of precursor molecules available for a more efficient diffusion into the polymer, can be delivered by modifying the conventional SIS process parameters. MDIS is a modified infiltration synthesis protocol which consists in repeating the precursor dosing multiple times while still maintaining static vacuum. The higher cumulative duration of precursor exposure in MDIS, when compared to conventional SIS protocol, determines a higher concentration of precursor molecules in the chamber. This translates into a higher number of molecules available to diffuse into the polymer, allowing the growth of a superior amount of material and a more uniform block-selective infiltration [31].

The control over precursors diffusion can also be exploited to expand the library of new multicomponent materials that can be synthesized with SIS. As recently reported by Azoulay et al., by designing the diffusive time of TMA and DEZ into cylinder-forming PS-*b*-PMMA, they were able to simultaneously grow different metal oxides at designated locations. Short TMA exposure times determined a shallow infiltration of the PMMA cylinder domains, whereas longer exposures of DEZ allowed a deeper diffusion into the entire film depth leading to the synthesis of an inorganic nanorod array of ${\mathrm{AlO}}_{x}-\mathrm{ZnO}$ heterostructures [51]. The full comprehension of the synthetic process requires also to consider the polymer–precursor interaction and its relation to the temperature, since their significant influence on the precursor effective diffusion. A clear insight into the role of temperature on the SIS was given by Weisbord et al. in a recent publication [52]. In a temperature-dependent model, the authors predicted the existence of a balance point temperature of thermodynamic equilibrium (ΔG=0) for each polymer–precursor pair. At the balance point temperature, the forward and reverse polymer–precursor interactions satisfy the thermodynamic conditions for maximum mass gain (Figure 4a,b).

The Lewis basicity of each polymer strongly influences the balance point temperature. For strong Lewis base polymers such as poly(2-vinylpyridine) (P2VP), high temperatures (≈$210\;{}^{\circ }C$) are desired for maximum mass gain. However, at these temperatures self-assembled BCPs such as PS-*b*-P2VP cannot maintain their pattern and consequently undergo morphology rearrangement that prevents the pattern quality of the infiltrated material. To overcome this issue, a multi-temperature SIS process was proposed. By the combination of a first low-temperature ( 80 ${}^{\circ }C$) SIS cycle followed by four SIS cycles at a higher temperature ( 150 ${}^{\circ }C$) the authors were able to obtain a higher mass gain for PS-*b*-P2VP when compared to single-temperature processes. Although being far from the thermodynamic conditions of maximum mass gain, the mass of ${\mathrm{AlO}}_{x}$ accumulated in the first SIS cycle at ( 80 ${}^{\circ }C$) prevents any BCP reconfiguration, preserving the vertically oriented cylinders pattern. Then, the subsequent high-temperature SIS cycles (150 °C) guarantee the highest mass growth (Figure 5).

## 3. Characterization Techniques

The development of the SIS process in terms of fabrication has progressed rapidly in the latest years, implementing a wide choice of materials for precursors and polymers and a large set of varying parameters regulating the infiltration process. However, the complete comprehension of the process mechanism and the exhaustive characterization of the materials’ properties have not yet followed through the expanding fabrication capabilities. Recent developments of lithographic, optical, mechanical and electrical applications of the SIS process require extensive characterization of the materials’ properties. A large set of physical and chemical methods has been applied so far with the aim to characterize the infiltrated polymeric nanostructures. The interest of the SIS community has been pointed at both the chemistry of reactions involved among the gaseous precursors and the polymer and the reconstruction of the morphology of the oxides nanostructures from a compositional and dimensional point of view. *In situ* characterization techniques have been used to unravel the phenomenology of the infiltration process inside the ALD chamber, while *ex situ* methods have been dedicated to the characterization of the results of the process at the end of different number of ALD cycles conducted under the same conditions. Given the wide variety of precursors and polymeric species used in literature and different process parameters, specific results of the characterization vary from study to study. Hereafter, we discuss how the different characterization techniques have been adopted for the study of SIS and we highlight the major achievements in understanding the process.

### 3.1. Phenomenology of the Infiltration Process

In the latest years, several *in situ* methodologies has been used and adapted inside the infiltration process chamber to gain direct access to the steps of the precursors infiltration in the polymeric matrix, i.e., the sorption of the gas-phase precursor, the diffusion and the entrapment inside the polymer [53].

Fourier-transform infrared spectroscopy (FTIR) is a well-known spectroscopic method based on the monitoring of adsorption peaks at different vibration frequencies in the mid-infrared range, constituting a fingerprint spectrum and corresponding to the chemical interactions among the reactants involved in a process. Integrated into the ALD chamber, FTIR is used for the temporal evolution analysis of the reactions between the organometallic precursor and the polymer functional groups at different stages of the ALD process. Transmission and reflection FTIR allow identifying the relevant moieties and the specific bonds that are formed (positive peaks) or consumed (negative peaks) or shifted in the phases of the infiltration process when changing the reaction parameters [35]. The spectral features are subtracted by a reference spectrum, acquired on a pristine substrate [27].

A notable example of the information retrieved from such spectral analysis is found in references [35,36], where some early results on *in situ* transmission FTIR measurements on PMMA thin films infiltrated with TMA were presented. The authors hypothesized and verified that the TMA reaction with PMMA occurs in a two-step process. The TMA is quickly absorbed by carbonyl C=O and ester C–O–R moieties in PMMA, forming a weakly-bound intermediate complex that is then slowly converted into a covalent bond, generating Al–O [35]. The analysis of temperature, thickness and time-dependence of the adsorption gave a deeper understanding of the process kinetics. The FTIR study highlighted that the adsorption of TMA into the PMMA film is a diffusion-limited process requiring long exposures to reach saturation with a quadratic functional dependence to time. The same time-dependence was observed in the desorption of TMA during purge time with desorption 10 times longer than adsorption [36].

Recently, another work on *in situ* FTIR measurements extended the analysis to different combinations of precursors (i.e., TMA and ${\mathrm{TiCL}}_{4}$) and polymers (i.e., PMMA, P2VP and PCL) to monitor the spectral changes of the reactive functional groups and kinetics of the adsorption and desorption processes [40]. Figure 6a,b report the absorption spectra of PCL acquired at the first and second SIS cycle at the two precursors’ exposure steps. Spectrum 3a.1 revealed a complete loss of C=O feature upon TMA interaction with the polymer, a blue-shift of C–O–R peak corresponding to a modification of the bond length and the formation of a ${\mathrm{AlCH}}_{3}$ complex. Upon the water dose, spectrum 3a.2 the C–O–R shift and aluminum complex peak are reversed indicating a loss of the surface species and complexed C–O–R. The C=O negative peak is not reversed indicating a unique irreversible covalent bond with TMA. Similar but less pronounced features are visible in spectrum 3b.1 corresponding to the first dose of ${\mathrm{TiCL}}_{4}$ in PCL. The spectrum presents C=O and C–O–R negative features, consistent with their consumption and a positive peak corresponding to the formation of a C–Cl complex. In this case, a non-covalent complex formation can be deduced from the spectrum 3b.2, where the reversed C=O peak suggests the partial release of these groups interacting with Ti–Cl species. For both graphs, the second SIS cycle is characterized by the same features, only with reduced intensities. The histogram in Figure 6c summarizes the FTIR results for the analyzed homopolymers reporting the percentage consumption of the reactive functional groups at different steps of the first SIS cycle for the two used precursors. This graph highlights the strong and stable reactivity of PCL to both TMA and ${\mathrm{TiCL}}_{4}$, allowing to identify PCL as a promising candidate for the infiltration process both as homopolymer and copolymer, matched with a non-reactive polymer such as PS.

Quartz crystal microbalance (QCM) gravimetry is quite often used in combination with *in situ* FTIR or alone to monitor the SIS process *in situ* [30,33,43,51,52,54]. It consists in employing a quartz crystal commonly used in deposition systems and modifying it with a thermally-equilibrated polymeric coating matching the polymer which is being infiltrated in the vacuum chamber of the ALD [54]. During the precursor adsorption and diffusion inside the polymer, the changes in the oscillation frequency of the quartz crystal are monitored and converted into the precursor mass uptake or loss of the polymer, through the knowledge of the material density and acoustic impedance. These features render QCM gravimetry a versatile technique, allowing to gain insights into the growth kinetics for every oxide in the SIS library [27,43,51] in both molecular layer deposition and etching processes [55].

The time-dependent measurements usually present an increase in the mass gain of the polymer during the exposure to the precursor, potentially reaching saturation with zero slope, followed by a mass loss in the purging step, when the unreacted reactants and byproducts are desorbed from the polymer. The slope of the mass gain in the different steps can provide information on the diffusivity of the precursors in the polymer. In Figure 7a, QCM gravimetric measurements are conducted on a PMMA thin film during the ${\mathrm{TiO}}_{2}$ SIS process using ${\mathrm{TiCL}}_{4}$ as precursor [33]. A large initial mass gain is displayed indicating a great diffusivity of the ${\mathrm{TiCL}}_{4}$ precursor in the polymer, followed by a modest rate of mass uptake in the following steps. The slope of the desorption step provides information on the kinetics of the process. The steep mass loss during the exposure to $H_{2}O$ vapor precursor in the ${\mathrm{TiO}}_{2}$ infiltration of PMMA suggests a fast kinetics between water and the ${\mathrm{TiCL}}_{4}$–PMMA complex and the release of different byproducts of such reaction [33]. Analogously, gravimetric measurements of the infiltration of two precursors, TMA for alumina and DEZ for zinc oxide growth, are reported in reference [51] for a self-assembled PS-*b*-PMMA film, revealing a much more abrupt and steep adsorption for TMA than for DEZ, thus indicating a faster diffusion for TMA. Gradual and long desorption of TMA from PMMA domains (not shown here) evidences a slow release of the organometallic precursor from the interaction with carbonyl groups in PMMA [27,35], as also highlighted with FTIR results.

The analysis of cycle-dependent net mass gain can be used to highlight differences in mass uptake under constant conditions. In the plot reported in Figure 7b for different polymers (PS, PMMA and PS-*b*-PMMA) a much smaller ${\mathrm{TiCL}}_{4}$ uptake was observed in PS compared to PMMA and PS-*b*-PMMA layers at the first cycle of the SIS process, due to the selective reaction of the precursors with PMMA carbonyl groups [33]. At the seventh cycle, a steeper decrease of the mass gain is observed in PMMA rather than nanostructured PS-*b*-PMMA layer imputable to the formation of a saturated layer and cross-linked polymer inhibiting further diffusion in the PMMA layer. This analysis allowed to hypothesize that the presence of inert polystyrene in the surrounding of the PMMA nanodomains allows channeling the diffusion of ${\mathrm{TiCL}}_{4}$ precursor to the PMMA available reactive sites.

Monitoring the results of a temperature-dependent QCM gravimetric analysis of the infiltration of TMA inside PMMA and P2VP homopolymers and BCPs films allowed the group of Segal-Peretz and coworkers to further shed light on the mechanism of the infiltration of TMA in reference [52]. The authors implemented a quantum-mechanical model to compute the changes in Gibbs free energy during the SIS growth and investigate the reversible bond formation for each precursor–polymer pair, predicting the specific temperature conditions at which the forward and reverse interaction occur at the same rate. Such thermal conditions promote the in-depth diffusivity of the TMA. Experimental verification through *in situ* monitoring of the mass gain in the predicted temperature range proved the validity of their model. The prediction and control of such important process parameters allowed the authors to grow alumina in P2VP self-assembled nanodomains, previously inaccessible, while preserving their morphology and maximizing the mass gain.

Spectroscopic ellipsometry (SE), commonly adopted in studying the dimensional and optical properties of thin films of various materials, consists of the measurement of the elliptical polarization state of a light beam reflected on single or stacked thin films, with the incident beam being linearly polarized. The incident and collection angle are set at the same value and the ellipsometry spectrum is modeled to determine up to two parameters at a time among refractive index, density, or thickness of the thin film. In the characterization of the SIS process, SE can be used to monitor the polymer modifications during the different steps of the infiltration process.

In reference [56], the authors reported time-dependent thickness and refractive index measurement for PMMA and PS film infiltrated with ${\mathrm{Al}}_{2}O_{3}$. The SE measurements (not shown here) indicate a strong swelling of PMMA during the first TMA diffusion, followed by a decrease in the purging step consistent with the out-diffusion of the physisorbed precursor. The following thickness increase is ascribed to the water dose and the formation of covalently bound Al–O species, already demonstrated in reference [36]. After the final purging step in the first cycle, the thickness of the polymer has increased with respect to its pristine state. After each of the following cycles the thickness slightly increased. The refractive index shows no significant variation after 10 cycles, the authors explained this by considering that the loading of ${\mathrm{Al}}_{2}O_{3}$, with higher n than PMMA, compensates for the density reduction due to swelling, leaving the refractive index substantially unaltered. The authors observed no significant variation of the PS thickness in the first cycle, but a slight increase after ten cycles, due to the absence of C=O reactive groups and to the cyclical loading and unloading of TMA in the film.

### 3.2. Characterization of the Infiltrated Materials’ Properties

After the infiltration is completed, *ex situ* characterization of the morphological and dimensional distribution of the oxide component inside the polymeric nanostructures is often carried with a plethora of methods, including several types of microscopic and spectroscopic techniques, gravimetry and mass spectrometry. Special attention is addressed at the diffusion of the gaseous precursors inside the polymer and in-depth distribution of the oxide growth.

Electron microscopy family includes several imaging techniques which use a high-energy electron beam to probe the surface or cross-section of a specimen. These include scanning electron microscopy (SEM), scanning transmission electron microscopy (STEM) and conventional transmission electron microscopy (TEM). These are by far the most commonly utilized techniques for the dimensional characterization of nanomaterials, requiring simple calibration of the magnification using calibration samples with features in the same dimensional range as the analyzed ones [57]. Electron microscopy has been widely reported for the morphological characterization of block copolymers nanopatterns or polymeric films treated with SIS of inorganic compounds [31,35,51,56,58,59,60,61]. Electron microscopy is often complemented by energy-dispersive X-ray (EDX) spectroscopy. It is based on the detection of characteristic X-rays produced from the interaction of high-energy electrons with the specimen atoms, allowing the univocal analysis of the elemental composition. This technique has been used for both in-plane and in-depth chemical characterization of the infiltrated polymer [31,43,51,59,61,62,63].

SEM enables the imaging of the topography of inorganic nanodomains in the BCPs template, through the collection of secondary electrons produced by scanning a focused beam of electrons on the surface. Detecting backscattered electrons adds information on the contrast among features with different elemental composition (Z-contrast) seen in the topographical image. This technique is broadly utilized since it does not require any peculiar preparation, except metallization on insulating specimens, and its interpretation is very straightforward.

TEM requires the transmission of the electron beam through the sample to form a high-resolution image. This technique requires quite long and destructive preparation to thin the sample below 100 nm, down to 5–20 nm, at which it is transparent to the incident electron beam and to mount it on a specific TEM grid. A common method to obtain a cross-sectional view of the sample is to cut lamellae using focused ion-beam (FIB) precision milling, while top-view TEM images can be obtained by detaching a thin layer of the specimen from the substrate. Figure 8a–d report TEM images of a thin BCPs template, constituted of a PMMA matrix embedding PS cylinders. The BCPs nanopattern was treated with 3 cycles (Figure 8a,b) or 10 cycles (Figure 8c,d) of SIS to infiltrate ${\mathrm{In}}_{2}O_{3}$, with trimethylindium (TMIn) and water as vapor precursors, and then annealed to remove the polymeric component leaving its inorganic replica [64]. The indium oxide is infiltrated preferentially in the PMMA matrix as revealed by the mesoporous structures in the figures. TEM enabled the measurement of the average size of the indium oxide nanocrystals up to (5.8±0.9) nm after 3 cycles and up to (11.8±1.4) nm after 10 cycles with reduction of the pore diameter. Moreover, comparing the TEM images of as-grown inorganic layers (images not shown) and after the annealing allowed the authors to investigate the structural modification of the inorganic template from amorphous ${\mathrm{InO}}_{x}H_{y}$ to ${\mathrm{In}}_{2}O_{3}$ with cubic crystalline phase, identified by measuring the lattice spacing. TEM is usually coupled with EDX for compositional analysis and fast Fourier transform (FFT) for structural analysis on the nanocrystals [31].

STEM is a variation of conventional TEM in which a focused electron beam is raster scanned across the sample, previously thinned to allow transmission. Several detection modes are available giving STEM great versatility. On-axis detection of transmitted electrons yields bright-field intensity imaging, while the detection of fore scattered electrons complements it with annular dark-field (ADF) imaging, or high-angle annular dark-field (HAADF) imaging, giving Z-contrast information. Reference [51] reports the realization of heterostructure nanorod array through the simultaneous and spatially-controlled growth of ${\mathrm{Al}}_{2}O_{3}$ and ZnO with a single SIS process in a BCPs film of PMMA cylinders in a PS matrix. HAADF STEM micrographs of the heterostructures acquired at different tilting angles are presented by the authors, showing contrast variation along the nanorods’ length. EDX maps revealed the distribution of the target elements, Al and Zn, mainly at the top and bottom part of a nanostructure, respectively. In the same manuscript the authors also report a cross-sectional 3D reconstruction of the heterostructures, obtained by EDX-STEM tomography. Recently, HAAFD-STEM imaging was used to resolve the infiltrated ZnO at the junction of vertical and horizontal PLA in a three-dimensional structure of poly(1,1-dimethyl silacyclobutane-*b*-styrene-*b*-lactide) (PDMSB-*b*-PS-*b*-PLA) triblock terpolymer with PS and PLA blocks domains [61].

Atomic force microscopy (AFM) and, more generally, scanning probe microscopy (SPM) are microscopic methods for the topographic characterization of films and nanopatterned materials. The use of a scanning probe allows mapping the surface of the specimen with lateral and vertical resolution in the nanometer range. The characterization of polymers treated with SIS has been dedicated to monitoring the morphological evolution before and after the infiltration at different cycles, mostly on resist films treated for increased etch resistance in lithographic processes [65,66]. These measurements usually highlight an increase of the lateral size of the nanostructure, with consecutive reduction of their pitch, up to their complete merging, and rounded edges with increased number of cycles. Morphological analysis on self-assembled PMMA cylindrical nanodomains revealed swelling of the polymer and 25% increase in their lateral size after 5 SIS cycles, as reported in reference [67], consistently with SE observation in reference [56]. Additionally, compositional information may be retrieved from phase signal in tapping mode AFM measurements and nanomechanical properties may be investigated through force-distance measurements. Reference [67] reports PeakForce tapping mode for quantitative nanomechanical mapping (QNM) on SIS-treated homopolymers and self-assembled block copolymers. Young’s modulus was monitored on the PMMA homopolymer layer and cylindrical nanodomains revealing an increased value after 5 and 11 SIS cycles, respectively, consistent with the incorporation of ${\mathrm{Al}}_{2}O_{3}$ inside the polymer and increased stiffness. The results are reported in Figure 9a. Force-distance measurements on PMMA exhibited a decrease in the adhesion after infiltration with respect to the pristine polymer, as shown in Figure 9b. The same measurements on PS revealed no change in the stiffness or adhesion forces of the polymer.

Time-of-flight secondary ions mass spectrometry (ToF-SIMS) is a destructive technique consisting in sputtering the material under study with a focused beam of primary ions. This generates secondary ions that pass through a time-of-flight mass spectrometer. When investigating polymeric samples, the use of bombarding ions clusters improves secondary ions yield and reduces damaging and molecular fragmentation [30]. The resulting composition, corresponding to different depths of the sputtering process and planar position of the rastering primary beam, is used to reconstruct the 3D cross section of the specimen, complementing the results from STEM and EDX spectroscopy. However, appropriate calibration standards are required for quantitative depth-profiling [68]. ToF-SIMS has been used to understand the depth distribution of oxides after SIS treatment, usually adopted in homopolymer layers such as PMMA [56,63] and PS [56], PET film and fibers [54], but also in block copolymers layers such as PS-*b*-P2VP both as micellar films [58] and self-assembled nanodomains infiltrated with ${\mathrm{SnO}}_{x}$ [43].

Thermogravimetric analysis (TGA), similarly to QCM gravimetry, yields information on the mass of infiltrated oxide in an *ex situ* process consisting in heating up the hybrid material and monitoring the weight change due to the loss of the polymeric volatile component. In reference [69], this technique confirmed the incorporation of alumina in polyethersulfone (PES) membrane with intact nanostructuration enabling the growth of laser-induced graphene (LIG). Among the techniques already mentioned for *in situ* phenomenological studies, FTIR and SE are also used in *ex situ* characterization. Attenuated total reflectance (ATR) FTIR, a variation of FTIR in reflection mode, has been reported in several works [62,69], including grazing incidence configuration [65], as a useful analytical method for cycle-dependent chemical characterization of the infiltrated polymer properties. Spectroscopic ellipsometry is often used in *ex situ* studies to measure the thickness variation of the polymer during the main processing steps (i.e., prior to SIS, after SIS, and after the polymer ashing) [62]. It can also provide information on the modified refractive index of the hybrid materials thus supporting application in optics and optoelectronics and related fields.

The characterization of the hybrid materials’ properties after the SIS process is supported by several methods described so far, dedicated to chemical, morphological, mechanical, structural and optical analysis. Some of the most common techniques, such as STEM and EDX and ToF-SIMS analysis, present time-consuming preparation or destructive operations, compromising the functionality of the investigated materials. Another category of analytical methods, not yet mentioned in this review, is constituted by X-ray techniques allowing non-destructive versatile multidimensional investigation at high-resolution in both laboratory settings and synchrotron facilities. Structural properties can be characterized through X-ray diffraction (XRD), where the peaks’ intensity and position in the diffraction pattern identify the atomic arrangement univocally yielding information on phase, crystallographic orientation, crystallinity, grain size, strain and defects. Local chemical and electronic structure around selected atomic species in a material can be retrieved through element-specific measurements of the first and second shell coordination distances by X-ray absorption spectroscopy (XAS) inner-shell methods. These rely on brilliant X-ray beams to probe the material with energies near the target element’s adsorption edge or far above it for near-edge X-ray absorption fine structure (NEXAFS), also known as XANES, or extended X-ray absorption fine structure (EXAFS), respectively. Finally, morphological properties at the nanoscale can be studied through X-ray scattering (XRS) methods that display scattered photon intensities as a function of the momentum transfer Q (1/Å). Particularly, GISAXS, operated in grazing-incidence mode and analyzing small-angle scattering, is not new to the BCPs community and has been largely applied to study the nanoscale morphology of BCPs templates [70,71].

A notable multidimensional *ex situ* characterization using the former methods has been recently presented in reference [64] to study the atomic-scale structure and the possible mechanism of nucleation of TMIn precursor in PS-*b*-PMMA BCPs. Powder diffraction (PXRD) analysis of the crystalline structure of as-grown hybrid ${\mathrm{InO}}_{x}H_{y}$/PMMA thin film, already shown in Figure 8a–d. The resulting XRD peaks exhibit high broadness indicating randomly distributed inorganic phase domains without long-range crystallographic order, compatible with an amorphous structure formed at low processing temperature (80 °C). Concurrently, EXAFS analysis was carried out on as-grown and annealed infiltrated PS-b-PMMA films showing a transition from ${\mathrm{InO}}_{x}H_{y}$ clusters to crystalline structures, whose local coordination environment after annealing was compatible with cubic ${\mathrm{In}}_{2}O_{3}$ and ${\mathrm{In}(\mathrm{OH})}_{3}$. In addition, high-energy X-ray scattering (HEXS) measurements have been paired with atomic pair distribution function analysis (PDF) and, in combination with EXAFS on annealed samples, confirmed the formation of an inorganic mesoporous film with sub-6 nm ${\mathrm{In}}_{2}O_{3}$ cubic nanocrystals. HEXS-PDF analysis allows to retrieve the size of the inorganic clusters at each SIS cycle as well as their possible atomic structures [64,72].

Another noteworthy X-ray analytical method is X-ray photoelectron spectroscopy (XPS), also known as electron spectroscopy for chemical analysis (ESCA), is a common technique for surface chemistry analysis, usually implemented with laboratory setup. An X-ray beam impinges on the sample surface and generates photoelectrons at different energies. The energy spectrum enables the identification of the surface composition, chemical and electronic state. The characterization of infiltrated polymers is usually carried out *ex situ* to determine the chemical state of the oxide growth or chemical modification of the infiltrated polymer [33,43,58,65,69,73]. In reference [43], the authors report XPS measurements on ${\mathrm{SnO}}_{x}$ infiltrated in P2VP homopolymer films as shown in Figure 10a,b. XPS enables the identification of Sn $3d_{5/2}$ and Sn $3d_{3/2}$ transitions visible in the spectrum, indicating that both tin oxides with Sn(IV) and Sn(II) oxidation states can be grown in the polymer layer. In reference [69], XPS was adopted as evidence of the incorporation of alumina inside PES membranes through the identification of Al 2p intense peak after the SIS process. Other works presenting X-ray-based characterization of the BCPs properties include reference [74] where GISAXS has been implemented to characterize the time-dependent morphological evolution of the BCPs matrix during the SIS process and reference [73] combining XPS with GISAXS and X-ray reflectivity (XRR) to study surface active polymer additives in BCP formulations.

With respect to other methods, such as FTIR, QCM or TEM analysis, characterization through the previous and other X-ray methods has not yet reached a widespread diffusion in the SIS community despite these could provide a better understanding of the process–structure correlation. The encouraging straightforward and non-destructive acquisition is still associated with some challenges with regards to separating the organic and inorganic contributions of SIS complex and hybrid structures in X-ray scattering, reflectivity and spectroscopic signals [27,64,72].

## 4. Control of the Materials’ Functional Properties by SIS

### 4.1. Optical Properties

The capability to selectively include metal oxide species inside self-assembled polymeric materials opened several opportunities in technological fields requiring the manipulation of light. A clear example is the realization of anti-reflective coatings (ARC) covering flat-panel displays of electronic devices, solar cells, curved optical elements or light-emitting diodes. To this goal, materials with refractive indices below 1.2 are required. To date, the literature describes two distinct approaches useful for the realization of BCPs-based ARC. The first approach relies on the increase of the absorption coefficient of incident light, occurring as a consequence of multiple reflections and scattering inside free-standing silicon nanopillars (SiNPs). In this context, the inclusion of metal oxides in ultrahigh molecular weight BCPs and the use of conventional reactive ion etching (RIE) processes allowed the formation of SiNPs with omnidirectional broadband anti-reflective capability (R < 0.16% in a wavelength range between 400 and 900 nm at an angle of incidence of 30°) [75]. A similar approach has been developed to obtain freestanding n-ZnO/p-Si nanotubes with low reflectivity in the UV-to-green light wavelength range (Figure 11a) [76]. The main drawback related to the use of SiNPs or nanotubes is the reduction of the transparency of the ARC film, strongly limiting the range of applications.

To extend the use of BCPs-based ARC to transparent substrates, the anti-reflective capabilities of an ARC can be tuned by adjusting its refractive index ($n_{\mathrm{ARC}}$) and thickness ($h_{\mathrm{ARC}}$), in such a way to induce destructive interference in the light reflected by the air/ARC and ARC/substrate interfaces. According to the Fresnel equations, for a given wavelength $\lambda_{0}$ and at a given angle of incidence, the best ARC conditions are accomplished for $n_{\mathrm{ARC}}=\sqrt{n_{\mathrm{sub}}\;\cdot \;n_{\mathrm{air}}}$ (being $n_{\mathrm{sub}}$ and $n_{\mathrm{air}}$ the refractive of the substrate and air respectively) and $h_{\mathrm{ARC}}\approx \lambda_{0}/4$, in the so-called “quarter-wave coatings”. Following this approach, Guldin and coworkers [77] realized one of the first examples of BCPs-based ARC, exploiting a combination of silica-based sol-gel chemistry and preformed ${\mathrm{TiO}}_{2}$ nanocrystals, selectively embedded inside poly(1,4-isoprene)-*block*-poly(ethylene oxide) (PI-*b*-PEO) micelles (Figure 11b). This type of composite materials combine the possibility of obtaining very low refractive index values (i.e., $n_{\mathrm{ARC}}=1.13$ at $\lambda_{0}=632$ nm) with self-cleaning properties. In fact, ${\mathrm{TiO}}_{2}$-based photocatalysis can be used to degrade the hydrocarbons adsorbed on the ARC and restore its pristine anti-reflective properties.

In 2017 Berman et al. [78] proposed a novel method, the solvent-assisted SIS, as an efficient approach to create conformal coatings with very low $n_{\mathrm{ARC}}$ (Figure 11c) over a broad spectral range. With this method, the refractive index of inorganic coatings can be finely tailored by tuning the geometric parameters of the BCPs template (i.e., film thickens, swelling ratio, porosity, feature size and periodicity) as well as the deposition parameters (i.e., type of infiltrated material, number of cycles). As a result, the authors demonstrated that the refractive index of ${\mathrm{Al}}_{2}O_{3}$ was lowered from 1.76 down to 1.10.

Beside the optical behavior linked to the change in refractive index, the nanostructured materials obtained by BCPs self-assembly and SIS exhibit interesting photoemission properties. Particular attention was paid to the electro- and photo-luminescence of nanostructures based on ZnO, a biocompatible and non-toxic material [79] with a wide range of potential applications in photonics [48,80], solid-state devices [81], gas sensors [82], water treatment [83] and biosensors [84].

The SIS process of zinc oxide is particularly complex, since the direct infiltration of diethylzinc (DEZ) precursor inside the polymer matrix often results in the formation of sparse ZnO nanoparticles [44,85]. For this reason a seeding treatment with a more reactive metal oxide (e.g., ${\mathrm{Al}}_{2}O_{3}$) is often required. Ocola et al. demonstrated that the seeding treatment and the polymeric matrix strongly influence the emissive properties of the ZnO nanostructures [41]. Figure 12a,d show the variation of PL spectrum at the earlier stages of the SIS (i.e., ${\mathrm{Al}}_{2}O_{3}$ seeding, first half cycle of DEZ, second half cycle of $H_{2}O$ and second half cycle of DEZ). Dimer Zn atoms (O–Zn–O–Zn and O–Zn–O–Zn–O) provide strong UV and VIS photoluminescence emission, 20 times greater than that obtained from the mono Zn atoms (O–Zn and O–Zn–O). For an increasing number of SIS cycles the authors observed the formation of crystals and consequent suppression of the VIS component of the PL emission (Figure 12e). It is worth noting that in the infiltration process of ZnO inside the PMMA matrix, the polymer does not constitute a passive host matrix for the DEZ precursor, but actively contributes to the PL of the nanostructures. Evidence of energy transfer between the PMMA and ZnO were demonstrated, while micro- and nano-patterning of the PMMA allows the manipulation of the PL spectrum of the infiltrated ZnO [86]. The large variation of the luminescence spectrum of ZnO, as a function of the deposition parameters, type and shape of the host matrix, represents a strong limiting factor to its diffusion in photonic applications. In this context, the infiltration of ZnO inside self-assembled BCPs matrices represent a viable way to obtain well-defined and periodic arrays of nanoparticles or nanowires (NWs) with improved photoemission capabilities in terms of spectral shape and intensity. In particular localized defects in ZnO nanoparticles, randomly disposed by drop casting on a pre-patterned substrate, have been reported to be efficient electrically driven single photon sources, working at room temperature [87]. The deterministic positioning and reduction of the dispersion in size of ZnO nanoparticles, achieved by combining SIS and BCPs, allows for the integration in electro-optic devices, such as electrically driven optical resonators.

### 4.2. Electrical Properties

The ability to control the level of doping of inorganic semiconductor materials has always driven the development of electronics. The same concept holds for the development of organic electronics where tailoring the doping level of organic functional materials is a prerequisite to control their electrical properties. With the growing interest in organic materials for printed and flexible electronics, light-emitting diodes (OLEDs), thin-film transistors, photovoltaic cells and batteries [88,89,90,91,92,93], several techniques based on the insertion of inorganic materials into polymers has been developed for the fabrication hybrid organic–inorganic materials with tailored electrical properties. Many of these processes alter polymer conductivity by doping with inorganic protonic acids, organic acids, Lewis acids, alkali metal salts or transition metal salts. These processes usually rely on wet chemistry with inherent limitations related to the solubility, temperature and can affect the polymer morphology, structure and purity [94,95,96]. In this scenario, SIS represents a solvent-free viable alternative to control electrical characteristics of polymers since infiltrated organometallic precursors lead to chemical reactions in the polymer to form hybrid materials with modified electrical properties. In 2015, Yu et al. [97] demonstrated that SIS represents a versatile doping strategy for engineering electrical properties of several functional polymers including polydimethylsiloxane (PDMS), polyimide (Kapton) and PMMA. The electrical properties of these polymers were tuned by infiltrating ${\mathrm{AlO}}_{x}$ molecules by SIS with TMA as a precursor. In the case of PDMS and Kapton, that always presents a negatively charged surface when contacted to other materials, it was observed that the ${\mathrm{AlO}}_{x}$ doping can significantly reduce the electron affinity of polymers due to the strong tendency of ${\mathrm{AlO}}_{x}$ molecules of repulsing electrons. Instead, if the host polymer possesses a strong tendency to repulse electrons as the ${\mathrm{AlO}}_{x}$ doping, as the case of PMMA, the effect of ${\mathrm{AlO}}_{x}$ doping is to enhance the positive charge density. By exploiting the different electron affinities of undoped and doped polymers, authors demonstrated the realization of triboelectric nanogenerators (TENGs) to convert mechanical energy into electricity. It is important to remark that, in this case, SIS was exploited as a technique for tuning bulk electrical properties since the diffusion of TMA was observed to be of ≈3 μm.

Among organic semiconductors, polyaniline (PANI) with its highly conjugated π delocalized molecular backbone is one of the most prominent conductive polymers finding applications in energy storage/conversion, supercapacitors, rechargeable batteries, fuel cells and water hydrolysis [98]. For all these applications, controlling the conductivity of PANI plays a crucial role. Besides depending on different oxidation states of the polymer (fully reduced leucoemeraldine, half oxidized emeraldine base and fully oxidized pernigraniline states) [98], the PANI conductivity can be modified through SIS doping. In 2017, Wang et al. [99] reported doping of PANI with metal chlorides by considering ${\mathrm{MoCl}}_{5}$ and ${\mathrm{SnCl}}_{4}$ precursors. In particular, it was observed that the conductivity of the infiltrated polymer (measured by means of four-probe techniques to avoid the effect of contact resistances) is correlated to the number of infiltration cycles and can be enhanced by up to six orders of magnitude. In the case of PANI infiltrated with ${\mathrm{MoCl}}_{5}$, the highest conductivity of 2.93 ×$10^{-4}\;S\;{\mathrm{cm}}^{-1}$ was observed after 100 infiltration cycles while in the case of PANI infiltrated with ${\mathrm{SnCl}}_{4}$ the highest conductivity of $1.03\times 10^{-5}\;S\;{\mathrm{cm}}^{-1}$ was observed after 60 infiltration cycles (as a reference, untreated PANI shows conductivity $\leq 1\times 10^{-10}\;S\;{\mathrm{cm}}^{-1}$). Despite the conductivity of traditional HCl-doped PANI outperforms these results (doping with 1 M HCl results in conductivity of $8.23\times 10^{-2}\;S\;{\mathrm{cm}}^{-1}$), it was observed that metal chloride doped samples exhibited chemical stabilization, due to a much lower impact of the thermal treatments in vacuum on the doped polymer conductivity. In this case, the effect of doping was ascribed to the oxidation of the PANI and complexation of metal chlorides with the PANI nitrogen, with consequent enhancement of the electron mobility along the polymer chain.

A strong improvement of conductivity was reported also in the case of PANI infiltrated by ZnO using DEZ as a precursor, where mutual doping in between inorganic species and polymer constituents was achieved [96]. Indeed, in this case, the process was responsible for a reinforcement of the binding of ZnO to nitrogen of the polymer chain backbone inducing a Lewis-acid type of doping and, at the same time, for doping ZnO with nitrogen forming an interpenetrated network. As can be observed from Figure 13a, the number of infiltration cycles can be tuned to alter the PANI conductivity. In all cases, the conductivity is higher than the HCl-doped PANI. Also, since the exposure time is correlated with the infiltration depth, better conductivity performances were observed in the case of extended exposure times. Figure 13b reports conductivities of PANI doped with different infiltration parameters calculated from slopes of I–V characteristics. A maximum conductivity of $18.42\;S\;{\mathrm{cm}}^{-1}$ was observed in the case of 600 infiltration cycles and 120 s of exposure time. It is worth noticing that the conductivity of the hybrid PANI/ZnO is a result of a synergy in between the involved materials since the conductivity is beyond the additive contribution of individual components. Indeed, lower conductivities were observed in the case of ALD-deposited ZnO films (refer to the conductivity represented by the green box of Figure 13b, where PANI was coated with an ${\mathrm{Al}}_{2}O_{3}$ infiltration barrier before coating with an ALD-deposited ZnO). Similarly, W. Wang et al. [100] reported a VPI process to dope poly(3-hexyl)thiophene (P3HT) by means of the ${\mathrm{MoCl}}_{5}$ precursor. In this case, the incorporation of Mo into the bulk polymer resulted in an increase of conductivity up to five order of magnitudes (a maximum of $3.01\;S\;{\mathrm{cm}}^{-1}$ was observed in the case of 100 infiltration cycles). In this case, changes in electrical conductivities are ascribed to a *p*-type doping related to the formation of a Lewis acid–base adduct formation between P3HT and ${\mathrm{MoCl}}_{5}$, where P3HT acts as a Lewis base in conjunction with ${\mathrm{MoCl}}_{5}$. In this framework, SIS results to be a promising strategy for solvent-free doping of polymers, making possible a top-down strategy to tune the electrical characteristics of pre-manufactured organic materials that can be implemented in roll-to-roll production lines for more efficient device fabrication of organic electronic devices. As a perspective, by properly selecting proper doping precursors and by controlling the infiltration conditions, the SIS strategy can be further explored for engineering electrical properties of a wide range of electrically conductive organic materials, where electrical characterization can be combined with UV-Vis, Raman, FTIR, XPS and XRD characterizations to understand chemical/structural changes of the polymer leading to a modification of its conduction properties.

Infiltrated polymers can be exploited also for the realization of transparent and multifunctional sensors, as reported by Ocola et al. [101]. In particular, in their work it is reported that the SU-8 (usually employed as negative resist for lithographic purposes) infiltrated with ZnO can be exploited for the realization of highly sensitive UV sensors. However, a detailed understanding of the sensing mechanism relying on volume interactions of UV light with infiltrated polymers still needs further investigation. Also, SIS was demonstrated as a versatile technique for the realization of electrochemically stable conductive membranes. In their work, Bergsman et al. [69] reported that a SIS-based process enables the realization of conductive LIG coatings on porous polymer substrates. Indeed, the infiltration of PES membranes with alumina by using the TMA precursor is responsible for stabilization against deformation above the glass transition temperature of the polymer. This enables direct lasing of these polymeric membranes to form an LIG coating without affecting the membrane pore structure, allowing the realization of permeable conductive membranes (Figure 14a). Also, these membranes were observed to be electrochemically stable. The sheet resistance of SIS-treated LIG membranes evaluated by the Van der Pauw method was observed to be dependent on the laser power (Figure 14b) achieving the value of $\left(37.7\pm 0.7\right)\;\Omega \;{□}^{-1}$, a value that is comparable to the sheet resistance of carbon-nanotube (CNT) composite materials. Note that without SIS treatment lased membranes exhibited an order of magnitude higher sheet resistance.

The SIS technique was reported also as a versatile technique to grow semiconductive oxide thin films, as reported by Waldman et al. [49] that have synthesized ${\mathrm{In}}_{2}O_{3}$ as a transparent conductive metal oxide. In their work, a process for growing ${\mathrm{In}}_{2}O_{3}$ by using TMIn as a precursor and PMMA as substrate was established. After subsequent removal of PMMA and annealing at 400 °C, the remaining SIS-derived film exhibited typical electrical characteristics of undoped ${\mathrm{In}}_{2}O_{3}$ thin films, as revealed by Hall effect measurements. Besides thin films, Vapor-phase infiltration can be exploited also for the realization of nanostructures based on metal oxides for the realization of electronic devices. For this purpose, the polymeric matrix can be patterned before the infiltration process in order to control position and geometries of nanostructures. In this framework, electrical properties of ZnO wires realized by means of SIS were investigated by Nam et al. [102]. As schematized in Figure 15a, the realization of ZnO stripes was performed by patterning a SU-8 template, subsequently infiltrated by ZnO and then removed by oxygen plasma. The resulting ZnO nanowires with length of 5 μm and width of about 50 nm present a nanocrystalline structure with grain sizes smaller than 5 nm. Subsequently, these nanostructures were contacted by means of source and drain contacts (Ti/Au) to realize an NW field effect transistor (NW-FET) device, exploiting the ${\mathrm{SiO}}_{4}$ substrate as gate dielectric and Si as gate electrode (schematization in Figure 15b). Electrical characterization revealed that the ZnO NWs become semiconducting only after an annealing process at 500 °C for 10 min in hydrogen (4% $H_{2}$ with Ar balance) to increase carrier concentration. After that, the ZnO NW exhibited a *n*-type semiconducting behavior as can be observed from Figure 15c, where an increase of the gate voltage ($V_{G}$) resulted in an increase of the device conductivity. Similarly, an intrinsic *n*-type doping was reported in a wide range of ZnO nanostructures. It is worth noticing that a similar unintentional *n*-type doping was reported in a wide range of ZnO nanostructures and was ascribed to the presence of intrinsic defects and/or impurities that act as shallow donors [103]. Assuming the cylinder-on-plate model and by considering the transfer characteristics reported in Figure 15d, the carrier concentration was estimated to be at least $2.5\times 10^{19}\;{\mathrm{cm}}^{-3}$ while the electron mobility was estimated to be about $0.07\;{\mathrm{cm}}^{-3}$. It should be noticed that the here reported charge density results to be much larger than the charge density observed in the case of single-crystalline ZnO NWs grown with a bottom-up approach that was reported to be in the order of $10^{17}-10^{18}\;{\mathrm{cm}}^{-3}$ [104,105]. In order to achieve new insights into the electronic transport mechanism of ZnO NWs realized by means of SIS with a top-down approach and to compare results with single crystalline ZnO NWs realized with a bottom-up approach, temperature-dependent electrical characterizations are required.

Recently, it has also been demonstrated that SIS represents an inexpensive and scalable strategy for the realization of resistive switching memories (ReRAM) that is compatible with existing semiconductor nanofabrication methods and materials. Indeed, Chakrabatarti et al. [106] have shown that nanoporous ${\mathrm{AlO}}_{x}$ grown by infiltration of PMMA acts as a dielectric layer for ReRAM cells characterized by a high on/off ratio (>$10^{9}$), low switching voltages (about 600 mV), retention up to $10^{4}$ s and pulsed endurance up to 1 million cycles. These characteristics make these cells promising for memory and neuromorphic applications.

Metal-oxide thin film nanoarchitectures can be also realized by combining SIS with self-assembled BCPs patterning exploited to generate nanomorphologies. By exploiting a MDIS protocol in hierarchical BCPs thin films, Subramanian et al. [31] reported the realization of three-dimensional ZnO nanomesh. Electrical conductivity across the multilayered nanomesh was observed to depend on the number of patterned layers. If a sufficient number of layers is realized, geometrical 3D charge percolation conduction is established across overlapping and orthogonal staking of nanowire fingerprint layers. For this reason, these systems represent percolative conduction networks where conductivity can be controlled by properly tuning geometrical parameters of the metal-oxide nanostructures. As a perspective, nanoarchitectures with tailored conductance properties can be realized by exploiting and combining different BCPs patterning strategies.

## 5. Conclusions and Perspectives

In recent years, a rapid expansion in SIS processing parameters has occurred [27]. Diverse vapor phase reactant combinations, pulses duration, purge duration, temperature and number of cycles have been tested on diverse polymers functional groups and block copolymers with varying Flory-Huggins parameter and molecular weight. The process kinetics and hybrid materials’ properties have been probed through several analytical methods so far, constituting both a challenge and a push for progress. However, developing more and more reliable characterization methods is required to increase our knowledge and control capability on SIS when moving in the expanding process space. The basic metrological requirements must be met proceeding towards absolute quantitative methods and interlaboratory comparability. A great deal of information on the chemical and structural properties of SIS-processed BCPs is to be found in complementary approaches using *in situ* and *ex situ* optical, vibrational and X-ray spectroscopic methods in combination with more straightforward information from electron and scanning probe microscopy methods. The interpretation of characterization results may be supported through theoretical modeling and simulations, with density functional theory (DFT) being a prominent candidate to investigate the mechanism of chemical reactions and predict suitable conditions and reactants [52,107]. In this scenario, advancements in SIS are related to the development of a high throughput metrology at the nanoscale.

The correct interpretation of the chemical/physical mechanisms and precise characterization of the infiltrated BCPs are fundamental characteristics for the realization of photonic structures and electronic devices with improved functionalities. A clear example is the fabrication of nanostructured materials with non-linear optical properties (e.g., ZnO nanostructures) [108] or metamaterials (e.g., metal/dielectric hyperbolic metamaterials) [109]. Furthermore, advances in BCPs patterning and SIS techniques can be exploited for the realization of either electrodes and/or active materials of next-generation electronic devices to overcome obstacles of device downscaling and system integration. As an example, BCPs in conjunction with SIS can offer an efficient way for fabricating crossbar arrays of memristive devices for the realization of next-generation computing architectures for neuromorphic-type of data processing, in accordance with the roadmap on emerging hardware and technology for machine learning [20].

Artificial intelligence (AI) and machine learning techniques, already giving increasing contribution to the field of physical chemistry [110], can support experimental and theoretical work on SIS process parameters control and characterization [111] in order to design functional materials with tailorable properties to be exploited in optical, mechanical and electrical applications through a “materials by design” approach.

## Figures and Tables

**Figure 1 nanomaterials-11-00994-f001:**
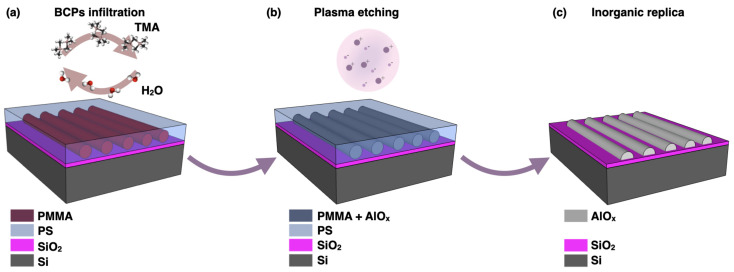
Schematic process flow of the sequential infiltration synthesis of block copolymers (BCPs). (**a**) ALD cycles with gaseous precursors (for instance trimethyl aluminum (TMA) and water). (**b**) Removal of the uninfiltrated polymeric component by plasma etching. (**c**) Inorganic replica of the BCPs template.

**Figure 2 nanomaterials-11-00994-f002:**
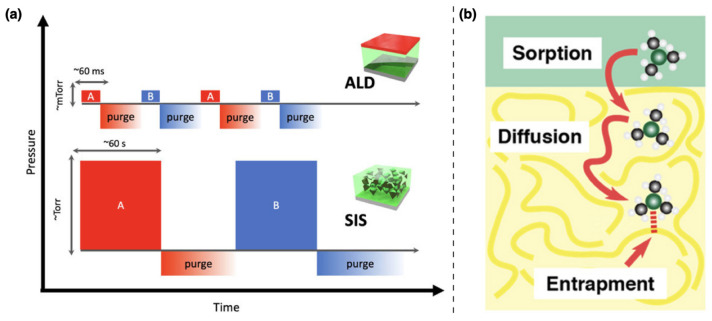
(**a**) Schematic comparison of conventional atomic layer deposition (ALD) and sequential infiltration synthesis (SIS) protocols. Reproduced and adapted from reference [27]. Copyright 2019, AIP Publishing. (**b**) Schematic illustration of metal–organic precursor infiltration process into polymers. Adapted with permission from reference [29]. Copyright 2019 American Chemical Society.

**Figure 3 nanomaterials-11-00994-f003:**
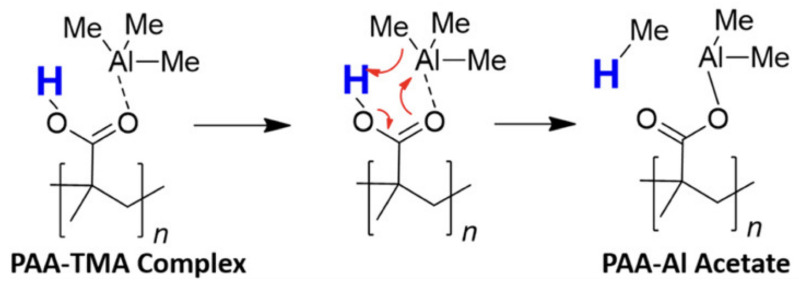
Proposed pericyclic mechanism for trimethylaluminum (TMA) and poly(acrylic acid) (PAA) reaction. Adapted with permission from reference [39]. Copyright 2019 American Chemical Society.

**Figure 4 nanomaterials-11-00994-f004:**
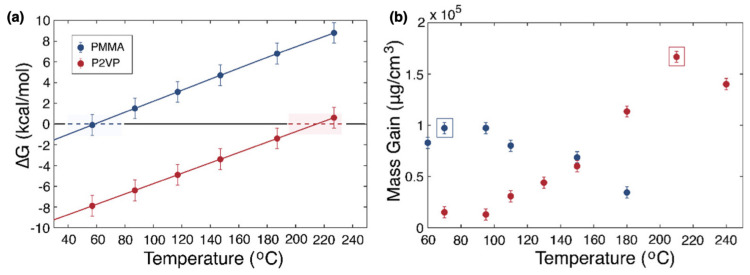
(**a**) Balance point temperature calculations for TMA-PMMA (poly(methyl methacrylate)) and TMA-P2VP (poly(2-vinylpyridine)) pairs and (**b**) relative experimental mass gain as a function of the temperature. Reproduced and adapted under the terms of Creative Commons Attribution 4.0 License from reference [52]. Copyright 2020 American Chemical Society.

**Figure 5 nanomaterials-11-00994-f005:**
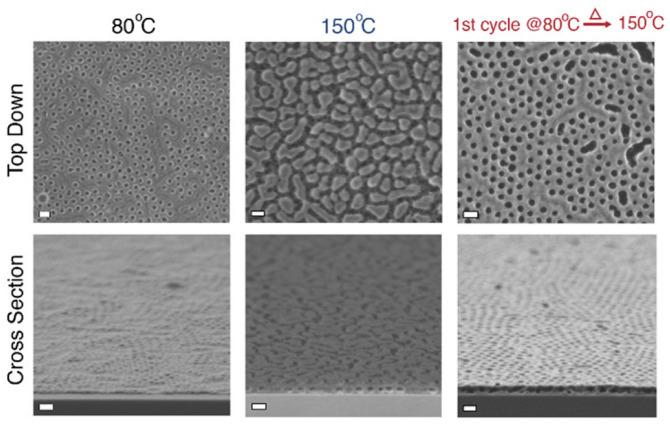
Top-down and cross-sectional scanning electron microscopy (SEM) images of ${\mathrm{AlO}}_{x}$ nanopatterns obtained after SIS at 80 ${}^{\circ }C$, 150 ${}^{\circ }C$ and multi-temperature processes. Scales bars are 100 nm. Reproduced and adapted under the terms of Creative Commons Attribution 4.0 License from reference [52]. Copyright 2020 American Chemical Society.

**Figure 6 nanomaterials-11-00994-f006:**
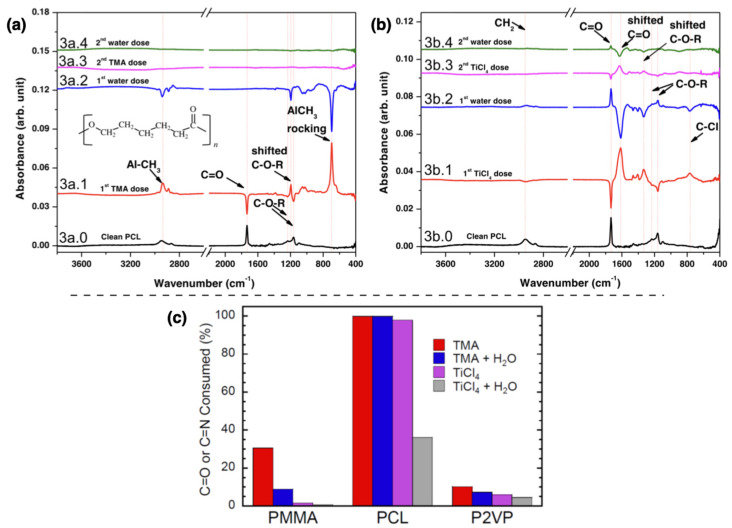
The adsorption spectra of poly(ϵ-caprolactone) (PCL) infiltrated with (**a**) TMA and (**b**) ${\mathrm{TiCL}}_{4}$ are shown. The spectra from bottom to top are referred to the pristine polymer layer (black line), the first SIS cycle (red and blue lines) and the second SIS cycle (pink and green lines). The histogram in panel (**c**) summarizes the percentage consumption of C=O (for PMMA and PCL) and C=N (for P2VP) functional groups at different stages of the infiltration process. All panels are reproduced and adapted with permission from reference [40]. Copyright 2020 American Chemical Society.

**Figure 7 nanomaterials-11-00994-f007:**
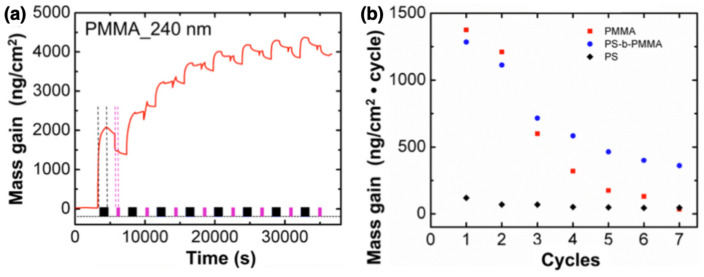
(**a**) Quartz crystal microbalance (QCM) gravimetry performed *in situ* during ${\mathrm{TiO}}_{2}$ SIS in a PMMA thin film. The graph displays the mass gain as a function of processing time. (**b**) Net mass gain on three different polymers (PS, PMMA and PS-*b*-PMMA) as a function of the cycle number. The graph is reproduced with permission from reference [33]. Copyright 2017 American Chemical Society.

**Figure 8 nanomaterials-11-00994-f008:**
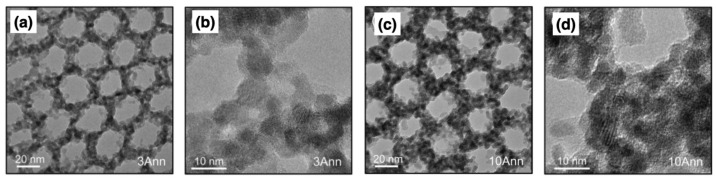
(**a**–**d**) TEM images at two different magnifications of the inorganic BCPs template, constituted of PS cylinders in a PMMA matrix infiltrated with (**a**,**b**) 3 cycles or (**c**,**d**) 10 cycles of ${\mathrm{In}}_{2}O_{3}$. (**a**–**d**) are reproduced with permission from reference [64]. Copyright 2019 American Chemical Society.

**Figure 9 nanomaterials-11-00994-f009:**
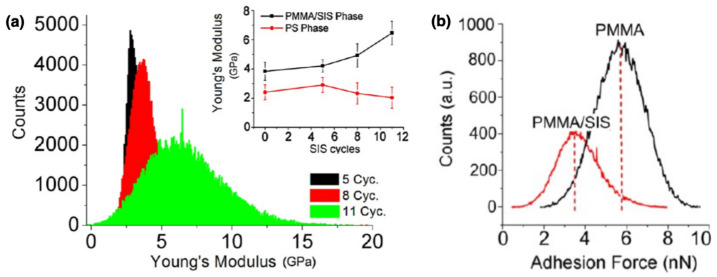
(**a**) Increase of the Young’s modulus variation at 5, 8 and 11 SIS cycles in PMMA domains. In the inset, the variation of the Young’s modulus for PMMA and PS phases is shown as a function of the number of SIS cycles. (**b**) Distribution of the adhesion force measured on PMMA nanodomains in before and after the infiltration process. All panels are reproduced with permission from reference [67]. Copyright 2017 American Chemical Society.

**Figure 10 nanomaterials-11-00994-f010:**
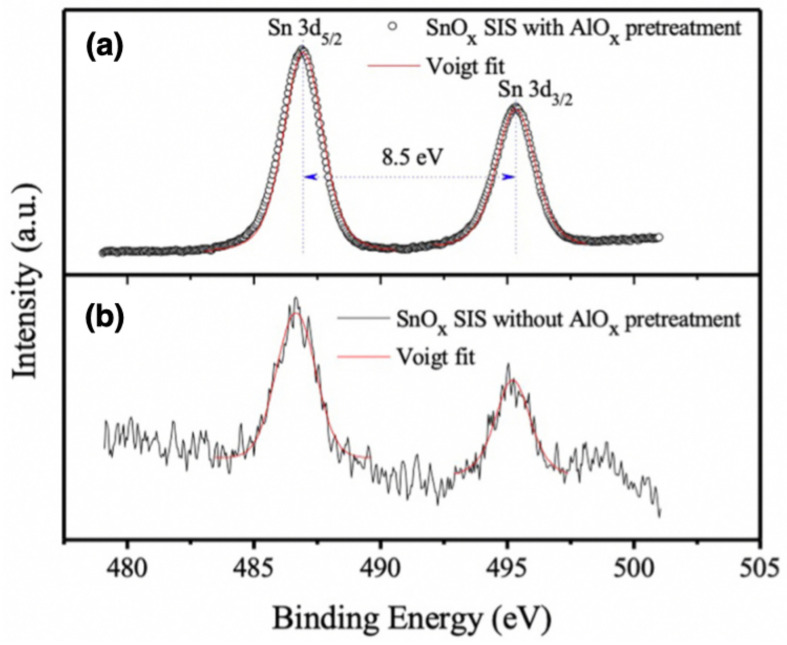
X-ray photoelectron spectroscopy (XPS) spectra recorded for Sn of ${\mathrm{SnO}}_{x}$ grown by SIS with (**a**) pre-treatment and (**b**) without pre-treatment processing showing both Sn $3d_{5/2}$ and Sn $3d_{3/2}$ (P transitions. Adapted with permission from reference [43]. Copyright 2019 Elsevier Inc.

**Figure 11 nanomaterials-11-00994-f011:**
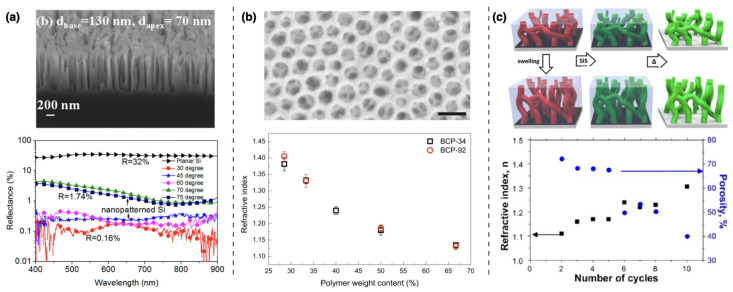
Broadband BCPs-based anti-reflective coatings (ARC) realized by (**a**) silicon nanopillars (**b**) ${\mathrm{TiO}}_{2}$ nanocrystals inclusion inside poly(1,4-isoprene)-*block*-poly(ethylene oxide) (PI-*b*-PEO) micelles and (**c**) sequential infiltration synthesis of ${\mathrm{Al}}_{2}O_{3}$ in cylindrical phase PS-*b*-P4VP. With these techniques, refractive index values approaching to $n_{\mathrm{ARC}}\approx 1.1$ can be achieved. (**a**) Adapted with permission from reference [75]. Copyright 2017 American Chemical Society. (**b**) Adapted with permission from reference [77]. Copyright 2013 American Chemical Society. (**c**) Adapted with permission from reference [78]. Copyright 2017 American Chemical Society.

**Figure 12 nanomaterials-11-00994-f012:**
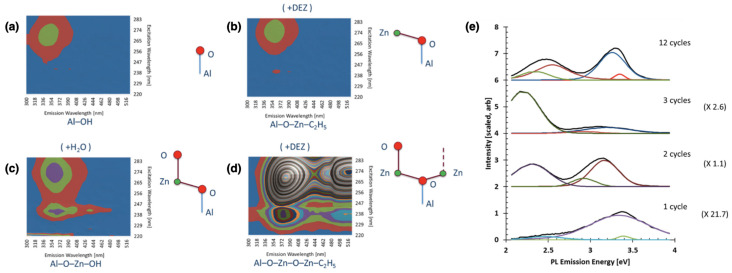
PL spectra recorded at different SIS steps and for variable excitation wavelengths between 220 and 285 nm: (**a**) Water terminated ${\mathrm{Al}}_{2}O_{3}$ seed layer, (**b**) first half cycle of DEZ, (**c**) second half cycle of $H_{2}O$ and (**d**) second half cycle of DEZ. The schematics on the right of all PL spectra illustrate the stage of ZnO growth that corresponds to each half cycle. (**e**) Emission spectra components as a function of the number of SIS cycles (the scaling factors are shown on the right side). All panels are adapted with permission from reference [41]. Copyright 2017 American Chemical Society.

**Figure 13 nanomaterials-11-00994-f013:**
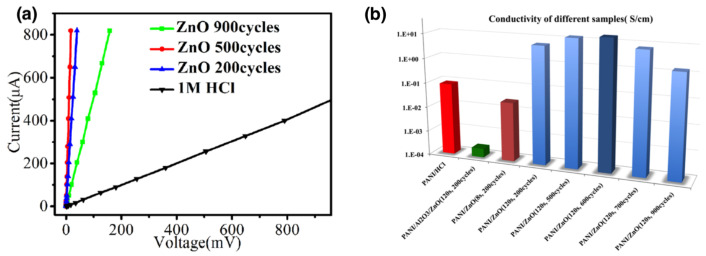
(**a**) I-V characteristics at room temperature of polyaniline (PANI) doped with different numbers of infiltration cycles (time exposure of 120 s). (**b**) Comparison of the conductivity of HCl-doped PANI (red box), atomic layer deposition (ALD)-deposited ZnO grown on PANI with an ${\mathrm{Al}}_{2}O_{3}$ infiltration barrier (green box) and PANI infiltrated with ZnO by using different exposure time and cycle numbers. All panels are adapted with permission from reference [96]. Copyright 2017, American Chemical Society.

**Figure 14 nanomaterials-11-00994-f014:**
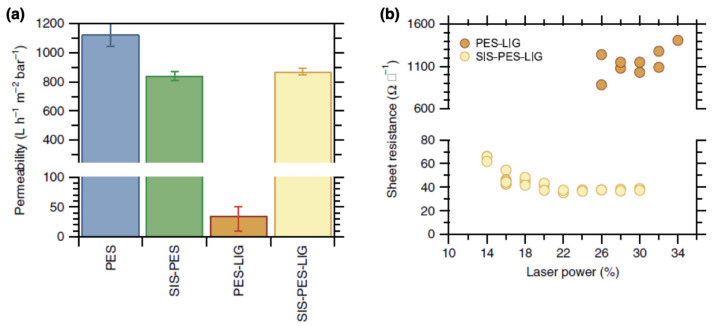
(**a**) Permeability of polyethersulfone (PES) membranes with and without SIS treatment before and after forming a laser-induced graphene (LIG) coating and (**b**) sheet resistance of lased membranes with and without SIS treatment as a function of the used laser power. All panels are reproduced and adapted under the terms of Creative Commons Attribution 4.0 License from reference [69]. Copyright 2020, the authors, published by Springer Nature.

**Figure 15 nanomaterials-11-00994-f015:**
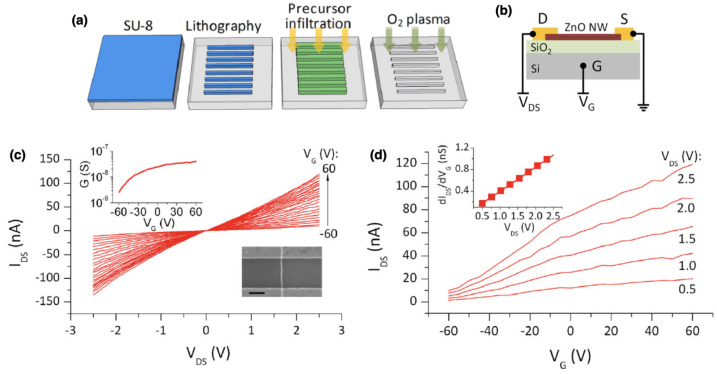
(**a**) Schematic representation of the ZnO patterning process consisting in the deposition of a SU-8 polymer, definition of SU-8 templates by lithography, infiltration synthesis with ZnO and formation of ZnO nanostructures by removing the polymer template through oxygen plasma. (**b**) Nanowire (NW) field effect transistor (NW-FET) transistor configuration where S, D and G represent source, drain and gate, respectively. (**c**) $I_{\mathrm{DS}}$ vs $V_{\mathrm{DS}}$ as a function of different $V_{G}$. The inset in the top left shows the dependence of the zero-bias conductance on $V_{G}$ while the inset in the bottom right shows an SEM image of the NW-FET (scale bar of 500 nm). (**d**) $I_{\mathrm{DS}}$ vs $V_{G}$ for different $V_{\mathrm{DS}}$. The inset shows the dependence of the transconductance (d $I_{\mathrm{DS}}$ / d $V_{G}$) on $V_{\mathrm{DS}}$. All panels are reproduced and adapted from reference [102]. Copyright 2015, AIP Publishing.

**Table 1 nanomaterials-11-00994-t001:** Polymers sorted by functional groups, utilized as templates for sequential infiltration synthesis (SIS) in the recent literature and the corresponding references.

Functional Groups	Polymers	References
Alkenes	PS-*b*-PI	[44,45]
Amides	PVP	[39]
Carboxylic acids	PAA	[39]
Esters	PS-*b*-PMMA PCL	[25,33,34,35,36] [40]
Epoxydes	PS-*b*-PIO	[44]
Pyridines	PS-*b*-P2VP PS-*b*-P4VP	[31,40] [46,47]

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
