# Peer review of "Recent Advances in Sequential Infiltration Synthesis (SIS) of Block Copolymers (BCPs)"

_nanomaterials, 2021, doi:10.3390/nano11040994_

Round 1

Reviewer 1 Report

The premise established is the need for large-scale nanopatterned materials and self-assembly of block copolymers pushed as the means for generating nanopattterns as templates for inorganic materials combined with SIS.  The authors do a very good job of describing simultaneously the need for such processing, the advantages rendered via SIS and the need for characterization. 

I do think that much of the characterization can be shortened substantially, particularly in cases, which is quite extensive in which the approaches are standard materials characterization and where there is not a novel finding or consideration that a reader may be surprised by.  As an example the description of standard approaches and unsurprising results of FTIR, XPS, etc (more below) are unecessary at the level described, these should be shortened if there is not anything of surprise.  As a contrast, the optics and need to characterize are fascinating, and should be described as written.  The authors should differentiate between the typical and the novel more clearly. 

Overall, I enjoyed the manuscript, and would recommend for publishing, however to advise it be edited to better emphasize the importance of the non-conventional and severely shorten the standard approaches prior to publication.  In my estimation, "pages" could be cut without sacrificing the manuscript and in fact this would improve the manuscript.  Specific comments below:

In abstract say they report a comprehensive description of the chemical and physical characterization techniques used for in situ studies of mechanism and ex situ measurements of resulting properties of infiltrated polymers- probably 5 nm patterning needed.  This concept is never discussed, where does this technology sit in terms of continued downscaling?  I'd like to hear more on this in regards to other standardized approaches of similar scale.  Where will we be gaining?

Figure 3 is provided with no insight.  For someone attempting to learn about this in a review, what is the main point besides you can do this to create various metal oxides.  What do the differences in structures indicate, what should I take from this?

Under section 2.2 the authors do a great job in terms of describing in a thorough manner most of the approaches however in discussing polar swelling solvents, this is very phenomenalogically described, its not clear how this gets to superior amount of material or more uniform block-selective infiltration, more descriptive text would help as is supplied  in therms of i.e. balance point temperature. 

Likewise, I’d like a more thorough description of the multi-temperature SS process, what is the mechanism at play here, particularly cycle to cycle at higher temp?

Characterization

FTIR, is this straight forward in terms of characterizing in situ, or is it tricky.   FTIR otherwise does not need such a lengthy discripter as it’s a very common straight forward technique, however, if there are difficulties in design for measurements, that should be described more thoroughly.  As the FTIR data presented is pretty clean, I assume this is a trivial measurement to take in this system?  If so, it does not need to be described in such detail, just shortly described as a useful approach with the ability to determine chemical process and formation as shown. 

I’d like to understand better the quantum mechanical model proposed and how the mass gain validated this model, that’s quite intriguing.    The QCM section however is quite good!

Overall, the in situ characterization section is quite good, just recommend cutting back as most techniques are standard to chemists and materials scientists alike.  I don't really need a text book lesson on every standard technique if noting unusual is done or found. 

Wonderful ex-situ characterization section, no changes needed through the first half through AFM.  When moving into XRD and XAX, XRS, it begins to read more as a list of techniques without the same specificity as to how that is specifically useful in characterization in these materials.  In other words, if I were studying such systems, I would immediately consider these avenues because this is pretty straightforward, but tell me something I may not know in terms of intriguing info I could render.  XPS is a great example of telling me the transitions I would see and showing me, more of that.  Otherwise, section could be considerably shortened as well.

Optical properties- this is much more intriguing than the state of the art spectroscopy characterization that is mostly understood.  The creation of new optic materials and a need to understand such systems is quite intriguing. 

Author Response

Reviewer 1

1. In abstract say they report a comprehensive description of the chemical and physical characterization techniques used for in situ studies of mechanism and ex situ measurements of resulting properties of infiltrated polymers- probably 5 nm patterning needed.  This concept is never discussed, where does this technology sit in terms of continued downscaling?  I'd like to hear more on this in regards to other standardized approaches of similar scale.  Where will we be gaining?

In the revised text, we revised the introduction section with a more in-depth assessment on how the nanopatterning by BCP self-assembly takes part in the framework of device downscale. Specific reference to a comparison between directed self-assembly (DSA) of BCPs to more traditional lithographic multi-step approaches has been made.

2. Figure 3 is provided with no insight.  For someone attempting to learn about this in a review, what is the main point besides you can do this to create various metal oxides.  What do the differences in structures indicate, what should I take from this?

We agree with the reviewer about the considerations. This figure has been removed since it was just a simple representation of the metal oxides obtained after direct infiltration of PS-b-PIO and did not provide any further insight on the mechanism or on the properties of such materials. Our intent was to illustrate to the readers some examples of different materials that can be synthesized by infiltration of polymers, such as PS-b-PIO, that are novel to SIS. To this end and according to the suggestion of Reviewer 3, we added a table in section 2.1 which gathers all the polymers discussed in the review sorted by their functional groups with relative references. We think that this would help the readers to have a clear overview of the different polymers that can be used in SIS and that would improve the overall quality of the review. 

3. Under section 2.2 the authors do a great job in terms of describing in a thorough manner most of the approaches however in discussing polar swelling solvents, this is very phenomenalogically described, its not clear how this gets to superior amount of material or more uniform block-selective infiltration, more descriptive text would help as is supplied  in therms of i.e. balance point temperature. 

To meet the reviewer suggestion, in section 2.2, we implemented the discussion of the swelling-assisted SIS, with a more in-depth description of the phenomena involved in the higher mass incorporation of swelled BCPs and in the MDIS protocol as well. 

4. Likewise, I’d like a more thorough description of the multi-temperature SS process, what is the mechanism at play here, particularly cycle to cycle at higher temp?

At the end of section 2.2, we included a discussion concerning the multi-temperature SIS process, giving more insights into the mechanism involved at different temperatures.

5. FTIR, is this straight forward in terms of characterizing in situ, or is it tricky.   FTIR otherwise does not need such a lengthy discripter as it’s a very common straight forward technique, however, if there are difficulties in design for measurements, that should be described more thoroughly.  As the FTIR data presented is pretty clean, I assume this is a trivial measurement to take in this system?  If so, it does not need to be described in such detail, just shortly described as a useful approach with the ability to determine chemical process and formation as shown. 

We thank the reviewer for her/his comment, we shortened the discussion on FTIR in the revised manuscript

6. I’d like to understand better the quantum mechanical model proposed and how the mass gain validated this model, that’s quite intriguing. The QCM section however is quite good!

The model implementation presented by Segal-Peretz et al. is fully detailed in the cited reference Weisbord, I.; Shomrat, N.; Azoulay, R.; Kaushansky, A.; Segal-Peretz, T. Understanding and Controlling Polymer-Organometallic Precursor Interactions in Sequential Infiltration Synthesis. Chem. Mater. 2020, 32, 4499–4508., however, we added a brief specification of the quantum mechanical model goal in the QCM gravimetry last paragraph.

7. Wonderful ex-situ characterization section, no changes needed through the first half through AFM.  When moving into XRD and XAX, XRS, it begins to read more as a list of techniques without the same specificity as to how that is specifically useful in characterization in these materials.  In other words, if I were studying such systems, I would immediately consider these avenues because this is pretty straightforward, but tell me something I may not know in terms of intriguing info I could render.  XPS is a great example of telling me the transitions I would see and showing me, more of that.  Otherwise, section could be considerably shortened as well.

This section was mostly revised to better highlight the interesting results in reference He, X.; Waldman, R.Z.; Mandia, D.J.; Jeon, N.; Zaluzec, N.J.; Borkiewicz, O.J.; Ruett, U.; Darling, S.B.; Martinson, A.B.; Tiede, D.M. Resolving the Atomic Structure of Sequential Infiltration Synthesis Derived Inorganic Clusters. ACS Nano 2020, 14, 14846–14860. while underlying the challenges associated with X-ray-based characterization, mostly related to the separation of the polymer (organic) and metal oxides (inorganic) contributions in the data analysis which can be quite challenging. 

Reviewer 2 Report

This review article by Cara, Lupi and coworkers highlights recent advances in the sequential infiltration synthesis (SIS) of block copolymers (BCPs). In particular, mechanistic considerations on SIS are first described and the role of various functional groups present in the BCPs' structures on the SIS process is also discussed. Then, the authors describe in detail both in situ and ex situ characterization techniques commonly utilized to understand both the SIS process and the properties of inorganic or hybrid organic/inorganic materials prepared by SIS. Characteristic examples directly related to the SIS of BCPs are given for each technique. Finally, the optical and electrical properties of developed nanodevices are presented and critically discussed.

Overall, the manuscript very well-written with a great level of English, well-organized with a clear thought progression by the authors, comprehensive (explaining adequately both basic concepts and more specialized terminology) and well-supported by the references provided. I believe that the article will be of great interest to the polymer/lithography community and the readership of Nanomaterials and, as such, I recommend its publication after correcting a few minor points (mainly associated with typos and grammar mistakes).

  • Could the authors construct or adapt a schematic from existing literature showing the whole process to make a nanodevice using SIS (BCP deposition, SIS, etching, etc...) and include it in the Introduction section? 
  • In Page 1, lines 23-27: The sentence "Although their technological...material deposition" took me 2-3 reads to fully understand what the authors want to express. Could you please rephrase the sentence or break into two smaller ones, so it's easier for readers to understand?
  • In Page 1, line 28: Please add "such" before "...as block copolymers (BCPs)..."
  • In Page 3, lines 75-79: Please include relevant references for this sentence.
  • In Page 3, line 102: This should better read " ...to polymers with amide and carboxylic acid functional groups, such as ...."
  • In Page 4, lines 130-131: This should better read "...of the mechanism involving the SIS with DEZ..." and "...pre-heating treatments play a key role in...".
  • In Page 6, lines 184-185: This should better read "...a first low-temperature (80 oC) SIS cycle..."
  • In Page 7, line 233: Please correct to "...PMMA occurs in a two-step process."
  • In Page 8, Figure 6 caption: The bond of C=O should be corrected to a double instead of single one.
  • In Page 9, line 307: This should better read "....the results of a temperature-dependent QCM gravimetric analysis..."
  • Please refrain from using plural for "microscopy" and "spectroscopy". Could the authors revise the manuscript appropriately in all cases plural form of the word "microscopy" or "spectroscopy" is used? As an alternative "microscopic/spectroscopic techniques" can be used instead.
  • In Page 11, line 366: I suggest changing the word "diffused" to "utilized"?
  • In Page 20, line 736: This should better read "Recently, it has also been demonstrated...."
  • In Page 20, line 740: the word "dialectric" should be changed to "dielectric"
  • Please provide the abbreviations for all journals provided in the References section where applicable and also update appropriately the volume and page numbers for References 7, 38, 49, 61, 102 and 103.

Author Response

Reviewer 2

1. Could the authors construct or adapt a schematic from existing literature showing the whole process to make a nanodevice using SIS (BCP deposition, SIS, etching, etc...) and include it in the Introduction section? 

We constructed a graphical scheme of the process flow of  sequential infiltration synthesis of BCPs.

2. In Page 1, lines 23-27: The sentence "Although their technological...material deposition" took me 2-3 reads to fully understand what the authors want to express. Could you please rephrase the sentence or break into two smaller ones, so it's easier for readers to understand?

We agree with the reviewer that the sentence was not clear. Accordingly, the sentence has been rephrased for a better understanding of the readers in “Over the last years, their technological improvements determined significant advances, approaching the fundamental requirements demanded by the continuous downscale of device features. However, conventional lithographic techniques are now facing their intrinsic technological and economic limits [10] in terms of large-scale pattern definition and material deposition.”

3. In Page 1, line 28: Please add "such" before "...as block copolymers (BCPs)..."

We included the suggested correction. 

4. In Page 3, lines 75-79: Please include relevant references for this sentence.

As suggested, the following relevant references have been cited at the beginning of section 2 :

  • Peng, Q.; Tseng, Y.C.; Darling, S.B.; Elam, J.W. A route to nanoscopic materials via sequential infiltration synthesis on block copolymer templates. ACS Nano 2011, 5, 4600–4606.
  • Peng, Q.; Tseng, Y.C.; Long, Y.; Mane, A.U.; DiDona, S.; Darling, S.B.; Elam, J.W. Effect of nanostructured domains in self-assembled block copolymer films on sequential infiltration synthesis. Langmuir 2017, 33, 13214–13223. 
  • Dandley, E.C.; Needham, C.D.; Williams, P.S.; Brozena, A.H.; Oldham, C.J.; Parsons, G.N. Temperature-dependent reaction between trimethylaluminum and poly(methyl methacrylate) during sequential vapor infiltration: Experimental and ab initio analysis. J. Mater. Chem. C 2014, 2, 9416–9424.

5. In Page 3, line 102: This should better read " ...to polymers with amide and carboxylic acid functional groups, such as ...."

6. In Page 4, lines 130-131: This should better read "...of the mechanism involving the SIS with DEZ..." and "...pre-heating treatments play a key role in...".

7. In Page 6, lines 184-185: This should better read "...a first low-temperature (80 oC) SIS cycle..."

8. In Page 7, line 233: Please correct to "...PMMA occurs in a two-step process."

9. In Page 8, Figure 6 caption: The bond of C=O should be corrected to a double instead of single one.

10. In Page 9, line 307: This should better read "....the results of a temperature-dependent QCM gravimetric analysis..."

11. Please refrain from using plural for "microscopy" and "spectroscopy". Could the authors revise the manuscript appropriately in all cases plural form of the word "microscopy" or "spectroscopy" is used? As an alternative "microscopic/spectroscopic techniques" can be used instead.

12. In Page 11, line 366: I suggest changing the word "diffused" to "utilized"?

13. In Page 20, line 736: This should better read "Recently, it has also been demonstrated...."

14. In Page 20, line 740: the word "dialectric" should be changed to "dielectric"

All suggestions from 2. to 14. have been addressed in the manuscript.

15. Please provide the abbreviations for all journals provided in the References section where applicable and also update appropriately the volume and page numbers for References 7, 38, 49, 61, 102 and 103.

We apologize for the inconvenience. All the journal names have been revised with the appropriate ISO4 abbreviations and all references have been fixed with the appropriate volume and page numbers. 

Reviewer 3 Report

In this review, the authors report the latest advances in nanostructured inorganic materials synthesized by infiltration of self-assembled block copolymers (BCPs), including the SIS mechanism, in situ and ex situ characterization techniques and the properties of the obtained materials. Overall, this is a detailed and thoroughly review, thus I recommend its publication in Nanomaterials after revision.

  1. It is recommended to make a Table gathering the references about different polymer templates with functional groups in section 2.1.
  2. The authors should check the references carefully, some page numbers are missing, for examples, ref. 38, 49, 61, 102, 103, etc.
  3. It is recommended to merge section 4. Optical properties and section 5. Electrical properties into one heading, similar with section 3.

Author Response

Reviewer 3

1. It is recommended to make a Table gathering the references about different polymer templates with functional groups in section 2.1.

We thank the reviewer for the recommendation. We added a table gathering the polymers discussed in section 2.1 used as templates for SIS, sorted by their functional groups and relative references.

2. The authors should check the references carefully, some page numbers are missing, for examples, ref. 38, 49, 61, 102, 103, etc.

We apologize for the inconvenience. All references have been fixed. 

3. It is recommended to merge section 4. Optical properties and section 5. Electrical properties into one heading, similar with section 3.

This recommendation was implemented in the revised version of the manuscript, a new section is called “Functional properties of infiltrated polymers”, splitted then in two subsection considering the optical and electrical properties.